# Using Recurrent Neural Networks to Compare Movement Patterns in ADHD and Normally Developing Children Based on Acceleration Signals from the Wrist and Ankle

**DOI:** 10.3390/s19132935

**Published:** 2019-07-03

**Authors:** Mario Muñoz-Organero, Lauren Powell, Ben Heller, Val Harpin, Jack Parker

**Affiliations:** 1Telematics Engineering Department, UC3M-BS Institute of Financial Big Data, Universidad Carlos III de Madrid, Av. Universidad, 28911 Leganes, Spain; 2School of Health and Related Research, University of Sheffield, Regent Court, S1 4DA Sheffield, UK; 3Centre for Sports Engineering Research, Sheffield Hallam University, S10 2LW Sheffield, UK; 4Ryegate Children’s Centre, S10 5DD Sheffield, UK

**Keywords:** ADHD, tri-axial accelerometers, deep learning, Recurrent Neural Networks (RNN), Long Short Term Memory (LSTM)

## Abstract

Attention deficit and hyperactivity disorder (ADHD) is a neurodevelopmental condition that affects, among other things, the movement patterns of children suffering it. Inattention, hyperactivity and impulsive behaviors, major symptoms characterizing ADHD, result not only in differences in the activity levels but also in the activity patterns themselves. This paper proposes and trains a Recurrent Neural Network (RNN) to characterize the moment patterns for normally developing children and uses the trained RNN in order to assess differences in the movement patterns from children with ADHD. Each child is monitored for 24 consecutive hours, in a normal school day, wearing 4 tri-axial accelerometers (one at each wrist and ankle). The results for both medicated and non-medicated children with ADHD, and for different activity levels are presented. While the movement patterns for non-medicated ADHD diagnosed participants showed higher differences as compared to those of normally developing participants, those differences were only statistically significant for medium intensity movements. On the other hand, the medicated ADHD participants showed statistically different behavior for low intensity movements.

## 1. Introduction

Attention deficit and hyperactivity disorder (ADHD) is a neurodevelopmental disorder affecting between 3 and 5% of children of school age [1,2]. ADHD symptoms such as inattention, impulsivity and hyperactivity profoundly affect the intellectual, emotional and social behavior of children suffering it [3] and have a relevant impact in their activity and movement patterns [4,5]. Children with ADHD tend to exhibit characteristic traits such as higher activity levels and fidgeting behaviors [6]. These symptoms have an impact on the movement patterns with particular connotations for each particular patient which are also affected by the intake or not of medicines for the ADHD treatment. The diagnosis of the ADHD disorder (according to the International Statistical Classification of Diseases and Related Health Problems (ICD 10)) requires the observation of certain symptoms in all three dimensions (inattention, hyperactivity, and impulsivity [7]) in more than one setting. The Diagnostic and Statistical Manual of Mental Disorders (DSM-IV [8]) and more recent DSM-5 [6] proposes a diagnosis of ADHD based only on the symptoms from two dimensions (with hyperactivity and impulsivity symptoms included in the same dimension [9]). This means that for a patient to be given a diagnosis of ADHD, hyperactivity must be present in more than one setting. However, current assessment procedures involve subjective observations and accounts of hyperactivity which may not be accurate.

In recent years, several studies have proposed the use of different sensors in order to assess the effect of ADHD symptoms over different physiological signals. The authors in [10] used an infrared motion-tracking system to measure motor activity. Other studies such as [11] have used electroencephalography (EEG) sensors in order to measure event-related potentials (ERP) to distinguish ADHD patients from normal participants. More recent studies, such as [4,5,12], have analyzed the use of wearable inertial sensors such as accelerometers and gyroscopes in order to assess the differences between normally developing and children with ADHD. Wearable inertial sensors are easy to wear without interfering with the participant daily activities and do not require the continuous monitoring of specialized staff. They can be used to record user movement-related signals over several hours or days, in particular if low battery consuming sensors such as accelerometers are used. Importantly, they can measure activity in a real-world setting adding ecological validity.

In this paper, a novel mechanism to assess the movement patterns for both medicated and non-medicated children with ADHD, as compared to typically developing non-ADHD controls, is proposed by using a Recurrent Neural Network and a similarity measure based on the Pearson Correlation Coefficient. A total of 36 children and young people (ages from 6 to 16) wore 4 tri-axial accelerometer devices (one at each wrist and ankle) during 24 consecutive hours. Half of the participants (18) were ADHD diagnosed children and young adolescents, and half were typically developing non-ADHD controls. Half of the ADHD diagnosed participants (9 in total) were taking medicines while the data was recorded, and the other half did not take any medicine during the 24-hour recording (nor the day before). Of the 18 controls, half of them were used to train a Recurrent Neural Network (RNN) and the rest of the participants (the other 9 controls, the 9 medicated and the 9 non-medicated ADHD diagnosed children and young adolescents) were used to assess their similarity to the group of controls in the training phase. The proposed assessment method uses a Recurrent Neural Network (RNN) to learn the internal dependencies and patterns from the time series form the 9 typically developing participants used to train it. Training the RNN with acceleration data from typically developing children will allow it to learn some common traits about how normally developing children move. Using the trained RNN for new participants will detect segments in the time sequences which are badly characterized by the RNN. Due to the high complexity of the different human movements, and the high number of possible human activities, the training data will not be able to encompass all the different patterns in normally developing children and young adolescents, but a subset of them. Therefore, there are going to be segments of data from all users which will not be properly characterized by the trained RNN. However, the number of badly characterized segments is expected to be higher for participants with ADHD. This paper presents both the RNN-based architecture and the results both for non-medicated as well as for medicated children and young adolescents with acceleration data in a normal school day.

This paper is divided into 5 sections. The first section presents the introduction and motivation for the research in this paper. The second section is dedicated to capturing the related previous research studies. The third section captures the materials and methods, presenting the details for the hardware used, the way in which participant were recruited, the demographic data for the participants in each group and the proposed algorithm and method used to obtain the results. Section 4 is dedicated to showing the main results obtained, and Section 5 summarizes the major conclusions.

## 2. Related Work

The use of wearable sensors in general, and accelerometer sensors in particular, has motivated the research in new mechanisms, methods and tools to support both the diagnosis and monitoring of children with ADHD. Accelerometer sensors are able to capture movement related time series from different parts of the body and serve as a valuable information source not only to characterize the ADHD particular effects but also to monitor the impact and behavioral changes of some interventions. The authors in [13], for example, examined how a particular intervention (using fidget spinners) affected children with ADHD’s gross motor activity and attentional functioning in class by making them wear accelerometers (placed in children’s hip area) during the day. The research in [14] used a wearable wristband with an accelerometer to catch the movement information of the user and provide visual feedback about the presence of ADHD related symptoms. Other recent studies have used accelerometers for the assessment of tremor during relaxation and flexion and extension of the index finger [4] or to monitor the effect of ADHD on driving [15].

Previous studies using accelerometer sensors such as [4,5] have found statistically significant differences in several parameters measured from children with ADHD compared with non-ADHD normally developing controls. By analyzing the forces applied in the manipulation of an object and using several sensors to measure the movement rhythmicity, the authors in [4] found statistically significant differences between participants with ADHD and non-ADHD controls. Similarly, the authors in [5], using a tri-axial accelerometer worn on the non-dominant arm, measured movements while a group of children attended three different school classes: art, language and mathematics. The authors also found statistically significant differences between the way children with ADHD and typically developing non-ADHD controls moved their upper limbs. Several diagnosis support tools and methods have been proposed based on these differences. Using machine learning methods based on pattern recognition from wearable sensor data (obtained when performing a continuous performance test in controlled environments), the authors in [12] showed good results in the classification of ADHD patients and non-ADHD controls. The authors in [12] made use of two inertial measurement units (IMUs) incorporating a tri-axial accelerometer and a gyroscope, attached to the waist and ankle of the dominant leg. The measurements were taken during an approximately 1-hour visit to a psychiatric consultancy. A total of 668 different features, combining time and frequency domain calculations were used. The authors made use of a support vector machine (SVM) to achieve up to 95% for both accuracy and sensitivity. Recent studies such as [16] have shown that the classification results can be optimized by using deep learning-based methods. In fact, deep learning techniques have been successfully applied for the recognition of human activities and movements based on wearable and mobile sensor data [17], achieving better classification results as compared to previous shallow methods [18,19]. From the different deep learning algorithms, Recurrent Neural Networks (RNN) have shown to optimally adapt to process time series data [20,21].

In this paper, we propose a novel architecture based on the use of a 2-layer Recurrent Neural Network (RNN) to learn the movement patterns from normally developing children as observed from the combination of the time series of 4 tri-axial accelerometers (one at each wrist and ankle). The RNN will be configured to minimize the reconstruction error when predicting the next 2 second acceleration data for the 4 sensors based on the past 6 seconds of data. Once trained, the RNN will be fed with data from other participants in order to assess whether the RNN is able to recognize previously learnt patterns. The Pearson Correlation Coefficient will be used to assess the reconstruction similarity. The authors in [18] not only showed that deep RNNs achieved better results for Human Activity Recognition than other Deep Networks such as Convolutional Neural Networks (CNN) and other shallow methods such as Support Vector Machines (SVM) but also showed that the architecture of the RNN will have an impact in the results depending on the dataset. The number of layers and the size of the memory cells must consider not only the complexity of the activities to be recognized, but also the size of the dataset in order to avoid overfitting issues. The study in [18] achieved optimal results for RNN structures stacking 2, 3 or 4 layers depending on the dataset. The authors in [22] achieved optimal results when stacking 3 layers. The structure proposed in this paper is adapted to the dataset that we have generated in the experimental part of the study.

## 3. Materials and Methods

### 3.1. Participants and Demographics

#### 3.1.1. Inclusion Criteria

The participants in the experiment group were included in the study if they were children and/or young people with ages between 6 and 16 and they had already been diagnosed with ADHD. Their parents were asked to confirm that the children had been fully assessed and diagnosed by the Paediatric Neurodisability or Child and Adolescent Mental Health Services.

The inclusion criteria for participants in the control group were to be in a similar age range (6–16) and to not have been diagnosed of ADHD, and were not siblings of a child with diagnosed ADHD (since ADHD is highly heritable).

#### 3.1.2. Recruitment, Setting and Data Collection

Members of the research team conducting this study used a convenience sample of normally developing children in order to recruit participants using a snowball sampling method. For participants diagnosed with ADHD, children living in South Yorkshire (United Kingdom) were recruited via a community group for parents who were referred to these groups by their child’s clinician.

The members of the research team carried out visits to all the participants (both in the experiment and control groups) in their homes. An informed consent was provided by the parent/legal guardian to confirm their child’s participation in the study. Moreover, all the participating children also signed assent forms to agree to participate. The parents/legal guardians also had to complete a SNAP IV questionnaire to assess the current severity of their child’s hyperactive behaviors. The participants were placed a labelled sensor (embedded in a black sweat band, Figure 1 and Figure 2) on each wrist and ankle. The sensors were worn for 24 consecutive hours. Moreover, the parents were shown how to position the sensors in the wristband in case the sensor moved during the 24-hour recording. This was also required to help the participants to put the sensors back on after some situations in which they had to take them off such as for bathing and showering. Parents were also asked to write down if the sensors had been removed and for what reason. This would help to explain some data anomalies such as periods of complete inactivity.

#### 3.1.3. Demographics

To carry out the experiment, 36 children/young adolescents were recruited. Half of them (18) had been diagnosed with ADHD and the other half (18) were typically developing participants. In the group of 18 children/young adolescents diagnosed with ADHD, half of them (9) were not taking medication for their ADHD when the data was recorded, and the other half (9) were on medication. Table 1 and Table 2 show the demographic data for the children/young adolescents participating in the experiment. The average age for ADHD participants was 8.4 years and the standard deviation 2. The average age for selected typically developing children was 10.4 years and the standard deviation 2.9. All the 36 participants attended school on the day on which data was recorded and performed the regular activities of a normal day at school. The recordings were conducted in the same season of the year (between October and December) 2017.

A managing ADHD group in Sheffield was used to recruit the participants with ADHD. This group was only available to parents of children who had been diagnosed with ADHD. All the participants with ADHD had been previously assessed by the Paediatric Neurodisability or Child and Adolescent Mental Health Services in Sheffield Children’s NHS Foundation Trust. The assessment followed the NICE guidelines and the DSM 5 criteria using a standard assessment and diagnosis process. A SNAP symptom questionnaire was also completed by the participants’ parents, as previously mentioned. The SNAP questionnaire helped us to assess their current symptoms.

### 3.2. Ethics

The School of Health and Related Research (ScHARR) ethics committee at the University of Sheffield was responsible for the ethics approval (reference: 013209). To guarantee that all participants were able to understand the information provided to them, several participant information sheets were written for different age groups.

### 3.3. Sensors and Data Gathering

A set of 4 Runscribe™ inertial sensors (Scribe Labs, CA, USA) were used for each participant (Figure 1). A tri-axial accelerometer was included inside each sensor device. The sensors were placed in both wrists and ankles and were aligned to their long axis with limb. The sensors were placed within an elasticated ‘sweat’ band (Figure 2). The sampling rate for each sensor was set at 10 Hz in order to allow a 24-hour collection period without draining the batteries and the memory available in each sensor device. By using a 10 Hz sampling frequency, most of the movements will be adequately characterized, since the large majority of the power in voluntary human arm movements is below 2 Hz [23], although occasional low amplitude maximal speed movements may be up to 8.4 Hz [24]. The sensors were synchronized by putting them together and performing a simultaneous initialization shaking movement which could then be easily detected by a preprocessing software. The sensors’ measurement was in the +/-16 g range. This was enough to avoid saturation problems (as also confirmed by the manual inspection of the data collected). The participants were asked to wear the sensors for a 24-hour period, as previously captured. After this 24-hour data recording, the sensors were programmed to automatically stop collecting more data. In case children removed the sensors at night or while bathing, the parents helped them to put them back on. The inspection of the accelerometer traces allowed us to confirm adherence to the continuous wearing of the sensors. In case a discrepancy was found the collection was repeated. The data was downloaded via a cable when the sensors were returned after the 24-hour recording and the data has been analyzed following the procedures described in the following sections.

### 3.4. Data Preprocessing

To adapt the sensed raw data to the movement pattern characterization algorithm proposed in the next sub-section, a two-step preprocessing algorithm is used as captured in Figure 3. A first preprocessing step compensates for the different sensor orientations by projecting the acceleration data into a geo-referenced coordinate system, and a second step synchronizes the information from the 4 sensors into the same time sampling instants.

The data samples recorded by each sensor represents the raw acceleration projections on the sensor anchored coordinate system (x, y, z) of the user acceleration sampled every 100 ms as measured in the wrists and ankles of the participants. Since the sensors move with the user, the sensor anchored coordinate system makes it difficult to compare acceleration patterns among similar movements with different sensor orientations. A preprocessing algorithm is used in order to estimate the geo-referenced vertical acceleration values (aligned with the gravity force) from the acceleration vector in the sensor anchored coordinate system as captured in the left part of Figure 3. The method comprises the following steps:

Each tri-axial acceleration is sampled at a particular sampling frequency f. Each sample comprises 3 acceleration values (one per axis in the sensor coordinate system) for the combined acceleration mixing together the acceleration caused by the gravity force and the movement related acceleration. The sample i for each accelerometer can be represented as $\vec{a_{t}\left(i\right)}=\left(a_{tx}\left(i\right),a_{ty}\left(i\right),a_{tz}\left(i\right)\right)$. For the particular case of the sensor located in the left foot’s ankle, the acceleration components have been labelled $\left(lf_{x}\left(t_{i}\right),lf_{y}\left(t_{i}\right),lf_{z}\left(t_{i}\right)\right)$, using *lh* for the left hand worn accelerometer, *rf* for the right foot’s device and *rh* for the right hand’s acceleration values.

The gravity acceleration vector (the acceleration due to the gravity force) at sample number i in the sensor coordinate system could be estimated by using a moving average-based low-pass filter using the following equation (similar to the method proposed in [25]):$\;\vec{g{\left(i\right)}^{’}}=\frac{\sum_{j=i-\frac{N}{2}}^{j=i+\frac{N}{2}}\left(a_{tx}\left(j\right),a_{ty}\left(j\right),a_{tz}\left(j\right)\right)}{N+1}$, where N is the number of samples in the averaging window.

The acceleration caused by the movement of the user is estimated according to: $\vec{a_{m}\left(i\right)}=\vec{a_{t}\left(i\right)}-\vec{g{\left(i\right)}^{’}}$.

$\vec{a_{m}\left(i\right)}$ could then be divided into vertical and horizontal acceleration components following the equation: $\vec{a_{mv}\left(i\right)}=\;\frac{\vec{a_{m}\left(i\right)}\cdot \;\vec{g{\left(i\right)}^{’}}}{{∥\vec{g{\left(i\right)}^{’}}∥}^{2}}\vec{g{\left(i\right)}^{’}}$ & $\vec{a_{mh}\left(i\right)}=\vec{a_{m}\left(i\right)}-\vec{a_{mv}\left(i\right)}$.

The movement-related acceleration projection over the gravity vector $a_{v}=\frac{\vec{a_{m}\left(i\right)}\cdot \;\vec{g{\left(i\right)}^{’}}}{{∥\vec{g{\left(i\right)}^{’}}∥}^{2}}$ is a geo-referenced acceleration sequence compensating the sensor orientation variations over time.

The second preprocessing step in Figure 3 synchronizes the 4 geo-referenced acceleration signals. A fast shaking movement holding the 4 sensors together is performed at the beginning of each recording for each participant. This fast shaking movement is detected and processed by the following algorithm:Let the index *i* point to the beginning of the vertical acceleration signal.Remove $a_{v}\left(t_{i}\right)$ from the time sequence if $∥a_{v}\left(t_{i}\right)∥<th1$, where *th1* is a threshold value between the maximum acceleration values before the shaking movement and the maximum acceleration value inside the shaking movement.*i = i + 1* and repeat the step 2 until $∥a_{v}\left(t_{i}\right)∥\geq th$1.Remove $a_{v}\left(t_{i}\right)$ values and continue incrementing the value of the index *i* until the following condition is met $\frac{\sum_{j=i}^{j=i+N}a_{v}\left(t_{j}\right)}{N+1}\leq th2$, where *th2* is a second threshold small enough so that the average vertical acceleration values inside the shaking movement are higher than *th2*.

### 3.5. Proposed Algorithm

The proposed algorithm to learn the movement patterns from typically developing participants is captured in Figure 4. The circles in the lower part of the figure capture the recent past time values for the vertical acceleration signals from the 4 sensors and are fed into a 2-layer LSTM (Long Short Term Memory)-based RNN. The number of memory units in each layer may be made different. In our case, we used 50 memory units for the first layer LSTM cells and 20 memory units for the cells in the second layer. This architecture is adapted to the particular dataset generated for the current study. The number of memory cells in the first layer is selected based on the number of acceleration samples used as inputs and the number of the memory cells in the second layer is adapted to the length of the sequence to be reconstructed as an output for the similarity computation. Adding more layers or memory cells did not provide better results but created overfitting issues. The output of the second LSTM layer is fed into a fully connected layer which is trained to try to reconstruct the next values from the 4 vertical acceleration signals by minimizing the mean square error. In our case, we used the last 6 seconds of acceleration data in order to predict the next 2 seconds of data (one third of the input duration). Using a small prediction window will make the reconstruction errors small even for new movement patterns (not present in the training data). Making the prediction window large will increase the reconstruction errors even for common patterns present in the training data since the low correlation of far future data. Since the use of the mean square error will equally weight the reconstruction error from each data segment in the training set (as added to the total error to be minimized in the training phase), those data segments which are most frequently repeated in the acceleration sequences from the training users will have a bigger influence in the adaptation of the internal parameters of the algorithm in order to better predict the upcoming values for these sequences. These most common sequences will therefore capture the movement patterns better charactering the control users in the training of the algorithm.

Once the algorithm is trained with data from some typically developing participants, the algorithm is used to assess the similarity of new participants to those control children/young adolescents by using the trained structure to try to reconstruct the future acceleration values from the new users’ acceleration time series and comparing the similarity of the reconstructed segments with the real data values. The Pearson Correlation Coefficient is used to assess the similarity between the real and the estimated signals.

The structure of an LSTM cell in the first layer is captured in Figure 5.

The behavior of connections in Figure 5 is defined by the following equations:(1)$$f_{t}=\sigma \left(W_{f}x_{t}+U_{f}h_{t-1}+b_{f}\right)$$
(2)$$i_{t}=\sigma \left(W_{i}x_{t}+U_{i}h_{t-1}+b_{i}\right)$$
(3)$$o_{t}=\sigma \left(W_{o}x_{t}+U_{o}h_{t-1}+b_{o}\right)$$
(4)$$c_{t}=f_{t}\times c_{t-1}+i_{t}\times tanh\left(W_{c}x_{t}+U_{c}h_{t-1}+b_{c}\right)$$
(5)$$h_{t}=y_{t}=o_{t}\times tanh\left(c_{t}\right)$$
where
$x_{t}=\left(lf_{t},\;lh_{t},\;rf_{t},\;rh_{t}\right)$ is a 4-component vector (containing the synchronized vertical accelerations for each sensor at time *t*), the × symbol represents a bitwise multiplication and the σ symbol represents the sigmoid function. All the LSTM cells in each layer share the same weights ($W_{f}$, $U_{f}$, $W_{i}$, $U_{i}$, $W_{o}$, $U_{o}$, $W_{c}$, $U_{c}$) and biases ($b_{f}$, $b_{i}$, $b_{o}$, $b_{c}$). Two LSTM layers are stacked together to improve the learning capability of the algorithm connecting the output of the first LSTM layer cells as inputs for the second layer cells.

## 4. Results

The algorithm presented in Figure 4 was trained with the acceleration information from typically developing children with IDs: Control 3, Control 5, Control 6, Control 7, Control 11, Control 12, Control 13, Control 14 and Control 17. Once trained, the algorithm in Figure 4 was used to detect acceleration data fragments containing movements which are poorly predicted by the trained algorithm in the other 27 participants (9 typically developing, 9 non-medicated ADHD diagnosed, and 9 medicated ADHD diagnosed participants). The total number of non-similar fragments was counted per participant in order to assess whether there were statistically significant differences between ADHD non-medicated and normally developing participants and between ADHD medicated and normally developing participants. The similarity threshold, as captured in Figure 4, was set to 2 different values to assess the differences in results. A sub-division of the acceleration fragments according to their movement intensity was also performed in order to find whether low energy regions in the acceleration time patterns were more or less similar than regions containing more intense movements. The results were captured in different sub-sections.

### 4.1. Total Number of Non-Similar Fragments

The acceleration data from the 27 participants under study were divided into 8-second windows. The first 6 seconds from each window were fed into the trained algorithm in Figure 4 and the next 2 seconds were used to calculate the reconstruction similarity. The similarity threshold was set to th = 0.5, meaning that all movement fragments showing a Pearson Correlation Coefficient below that value w considered not to be present in the training data (or at least the RNN in Figure 4 was not able to learn that particular pattern from the data). The total number of non-similar detected fragments per group is captured in Table 3.

The results in Table 3 show that the average number of non-similar fragments in the 24-h data sequences for participants in the control group was smaller than those in the other 2 groups, and there was also a smaller deviation in the data among the participants in the control group. For ADHD diagnosed children/young adolescents, the number of non-similar movement fragments, as compared to the patterns learnt by the algorithm in Figure 4 trained with 9 participants in the control group, was bigger, showing a higher deviation among participants. It is also interesting to note that ADHD medicated participants moved in a more similar way to the control group.

To assess whether the differences between both ADHD diagnosed groups with the typically developing control group were statistically significant, the t-test was used. Table 4 shows the results for both groups. The results were not statistically significant under the 0.05 threshold value for any of the groups, although the value for the non-medicated group was close to the threshold. A further analysis dividing the acceleration fragments into similar intensity movement intervals was therefore required in order to assess for what particular types of movements the differences showed a higher significance.

Changing the parameters in the architecture presented in Section 3.5 (number of layers and memory cells per layer) provided similar results, but required more time for training and was prone to overfitting problems. A 3-layer RNN architecture with 50 memory cells in all 3 layers achieved a p-value of 0.054, but required 73% more time to train, and the difference in non-similar segments between the control group used for training and the control group used for validation was 52% (a greater number of parameters to be trained in the model would attempt to reduce the errors in the training set without guaranteeing generalization to other control participants not used in the training phase).

### 4.2. Interval Analysis

The data fragments for each participant were divided into different categories according to the intensity of the acceleration values as measured by the standard deviation of the combined data from all 4 sensors in the fragment (the 8-second window). For low-intensity movements, the standard deviations would be small, while for more energetic movements, the standard deviations for the combined acceleration from all 4 sensors in the time window would be higher. The algorithm in Figure 4 was also trained with data fragments from training users in the same intensity intervals. The results are illustrated in Figure 6 and Figure 7.

For non-medicated ADHD participants (Figure 6), the differences from the control group are statistically significant in the region from 0.4 to 1 for the standard deviation measured in g units (moderate-intensity movements). For low-intensity movements, as well as for high-intensity vertical acceleration data fragments, the ADHD non-medicated group showed more similar behavior when compared to the control group.

The results for medicated ADHD participants (Figure 7) show very different results. Their movement patterns show fewer differences from the movement of the control group, and the only statistically significant region corresponds to the low-energy movement interval.

The area under the curve for non-medicated ADHD diagnosed participants for movement fragments with the optimal combined standard deviation (between 0.8 and 1 in g units) is illustrated in Figure 8. The same curve for the second optimal interval is presented in Figure 9. In the optimal operation point, 7 out of 9 non-medicated ADHD diagnosed participants showed more badly classified acceleration fragments using the algorithm in Figure 4 than the threshold, while only 2 out of 9 normally developing participants executed more non-similar movement fragments than the threshold (getting a low similarity value from the proposal in Figure 4).

Figure 10 depicts the area under the curve for medicated children in the optimal interval for movement intensities (standard deviation between 0.2 and 0.4 in g units). In this case, 5 out of 9 medicated participants showed more non-similar movement fragments than all the normally developing participants in the control group (movements that the RNN structure in Figure 4 did not learn from the control participants used in the training phase for the same movement intensity intervals).

### 4.3. Comparing Similarity Threshold Values

To assess the effect of the similarity threshold value in the results, different thresholds were analyzed for each group and movement intensity interval. The results were similar for different values. Figure 11 illustrates the particular case of ADHD non-medicated children with movements in the interval of 0.4 to 0.6 g. The optimal threshold value in this case was th = 0.5 (which was the one used in the previous sub-sections), and the difference in the results was small.

## 5. Conclusions

This paper presents a novel architecture for characterizing movement patterns based on the use of a Recurrent Neural Network (RNN) using Long Short Term Memory (LSTM) cells. RNNs have previously been used in the characterization of time series showing optimal results for solving Human Activity Recognition (HAR) problems.

The RNN part of the proposed method was used to learn the movement patterns in the control group that is used to train the network. The training was based on learning the optimal patterns in the recent past information of the time series obtained from 4 different tri-axial accelerometers (one on each wrist and ankle) in order to optimally predict the information in the near future for the same signals. A mean square optimization is used to minimize the reconstruction errors. Once trained, the acceleration data from new users will be fed into the network and a similarity measure based on the Pearson Correlation Coefficient will assess whether the new information shows similar patterns to those found in the users when training the network.

For this research study, 36 recruited children were used. These children were divided into 4 groups:9 normally developing participants used for training the network only9 different normally developing participants used in the validation part of the study9 non-medicated ADHD diagnosed participants9 medicated ADHD diagnosed participants

A comparison under the null hypothesis “there is no difference in the mean values for poorly predicted acceleration fragments between the participants in groups 2 and 3 and between the participants in groups 2 and 4” was performed using the t-test. If the total number of non-similar acceleration fragments was computed for all movement intensities together, there was no statistically significant differences between non-medicated or medicated ADHD diagnosed participants and the users in the normally developing participant control group, under the 0.05 significance threshold (p-values lower than 0.05), although the non-medicated showed a more significant difference.

Analyzing the differences by dividing the different movements according to their intensity, as measured by the combined standard deviation from the vertical acceleration signals, there were some significant differences among groups. The non-medicated participants with ADHD showed statistically significant data differences for medium intensity movements. The learned movement patterns for the 9 training controls by the algorithm in Figure 4 were not able to capture the patterns in the ADHD non-medicated group in a significant higher number of times as compared to the other 9 participants in the control group (those not used for training) for values of the combined standard deviation for the vertical acceleration data from 0.4 to 1 in g units. For medicated participants, the movement patterns in the medium intensity segments were better predicted by the learnt patterns from the control training group. However, the movement patterns in the low energy interval (from 0.2 to 0.4 in g units) showed statistically significant differences as compared to the normally developing participants. One possible cause maybe the effect of medicines in lowering the movement intensity of some ADHD characteristic movements. Previous similar research studies, such as [5], have also shown that for some particular activities there are statistically significant differences in the regions of the moderate intensity movements (0.5–0.8 g in that particular case) for the upper limb movements, although the authors did not perform an analysis for non-medicated vs medicated participants. In particular, the study in [5] showed bigger statistical differences in the acceleration values of the upper limbs between children diagnosed with ADHD and normally developing children while attending an art class than for math’s or language classes. Our study extended the conducted activities to a complete 24-hour period and divided the participants diagnosed with ADHD into medicated and non-medicated cases. The authors in [5] only analyzed raw data values, while our study proposed a state-of-the-art architecture based on RNN to better capture time dependencies and patterns. Finally, the study in [5] only used 17 participants (10 diagnosed with ADHD, 7 normally developing cases) while we incorporated 36 participants (18 diagnosed with ADHD, 18 normally developing cases).

As future work, a deeper analysis of finding common patterns in ADHD diagnosed children will be performed. The effect of medication on behavioral changes over these patterns will also be studied. A more exhaustive experiment including a higher number of participants is also expected.

## Figures and Tables

**Figure 1 sensors-19-02935-f001:**
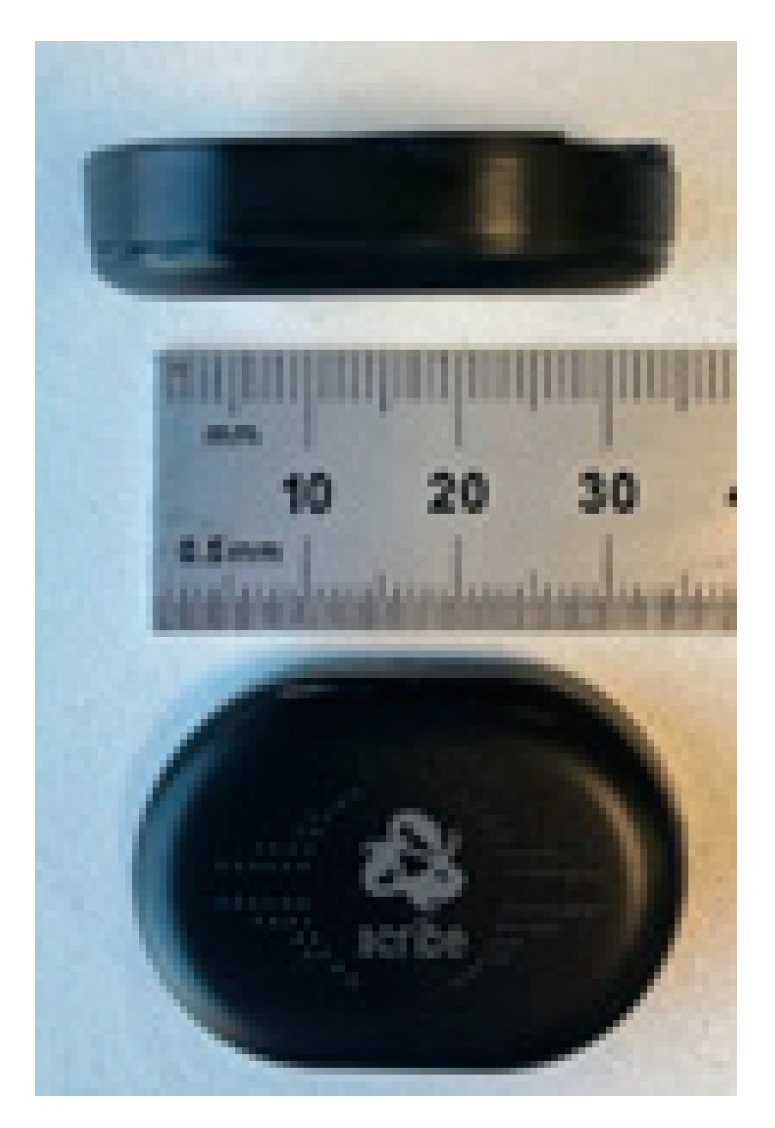
Sensors used.

**Figure 2 sensors-19-02935-f002:**
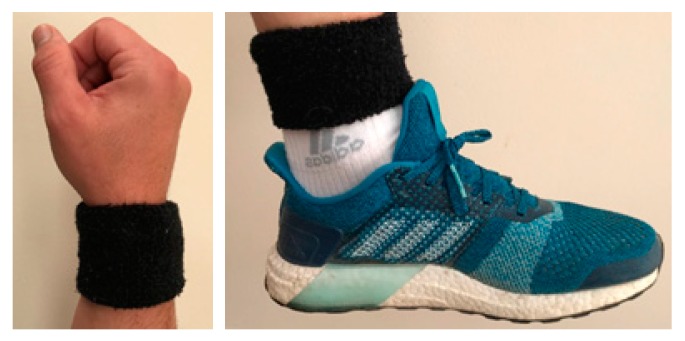
Positioning of the sensors.

**Figure 3 sensors-19-02935-f003:**
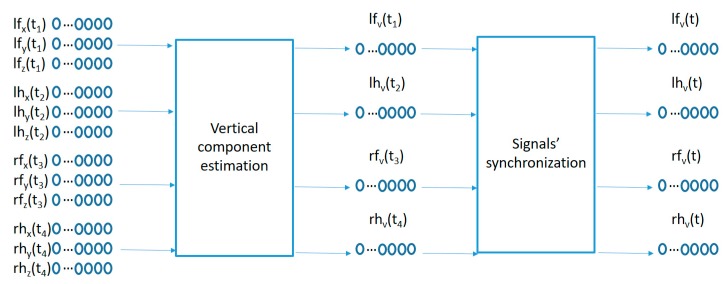
Signal preprocessing.

**Figure 4 sensors-19-02935-f004:**
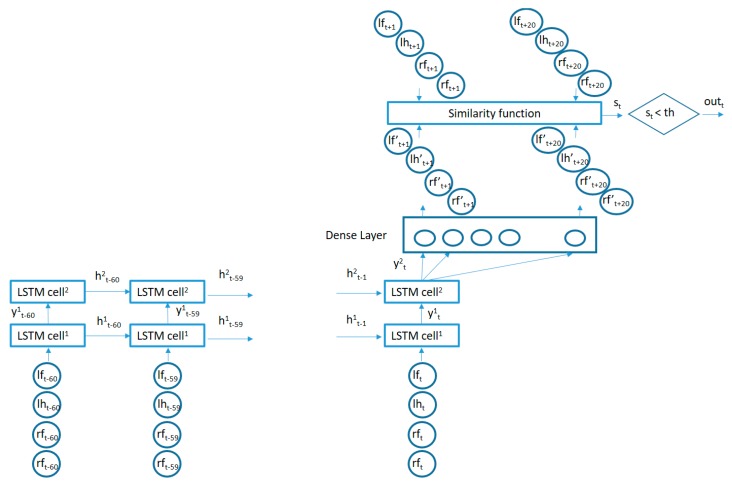
Proposed algorithm.

**Figure 5 sensors-19-02935-f005:**
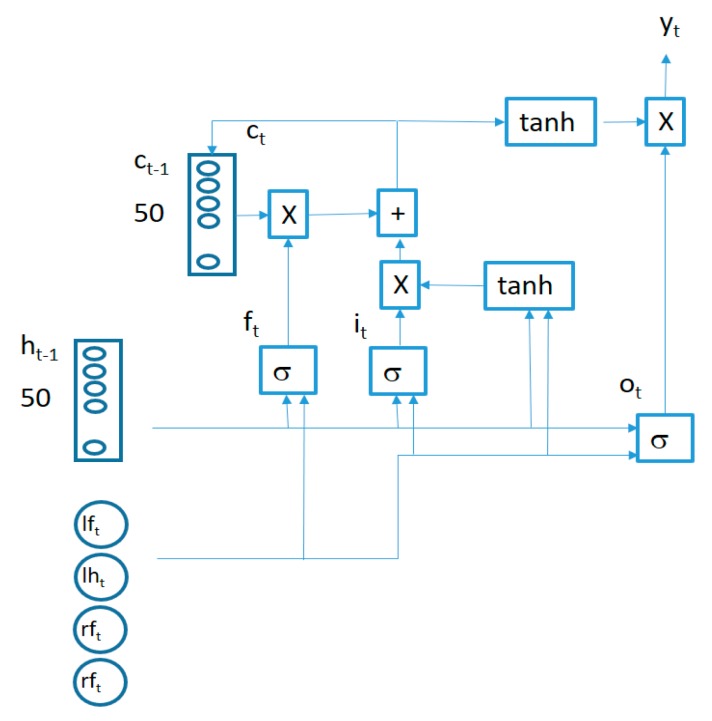
LSTM cell structure.

**Figure 6 sensors-19-02935-f006:**
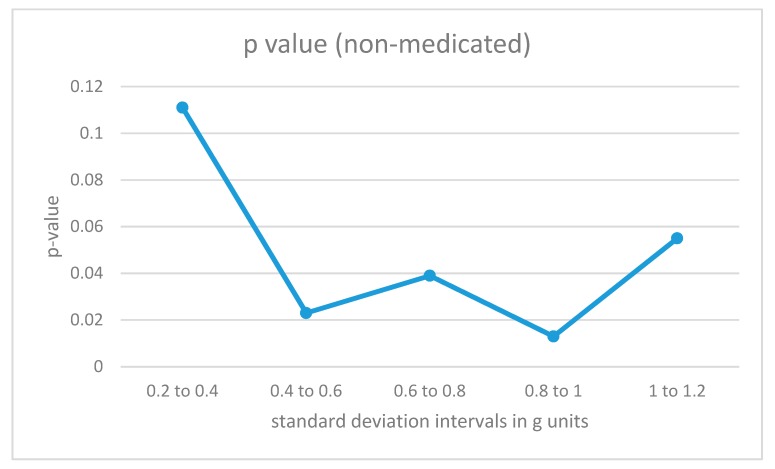
p-Values for acceleration intervals with different movement intensities for non-medicated participants with ADHD.

**Figure 7 sensors-19-02935-f007:**
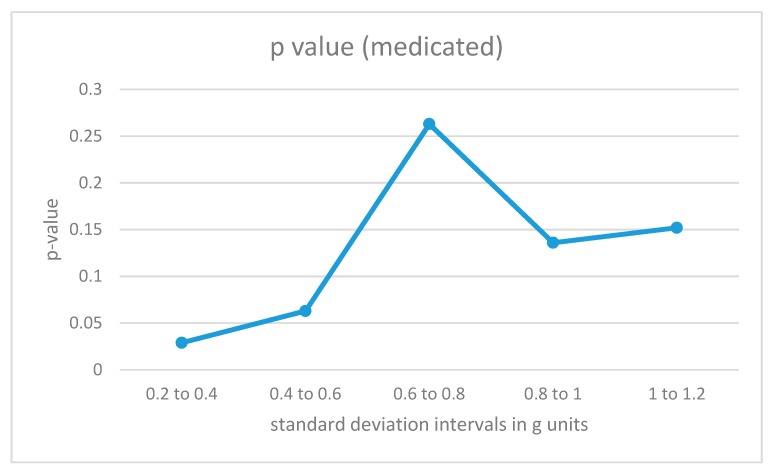
p-Values for acceleration intervals with different movement intensities for medicated participants with ADHD.

**Figure 8 sensors-19-02935-f008:**
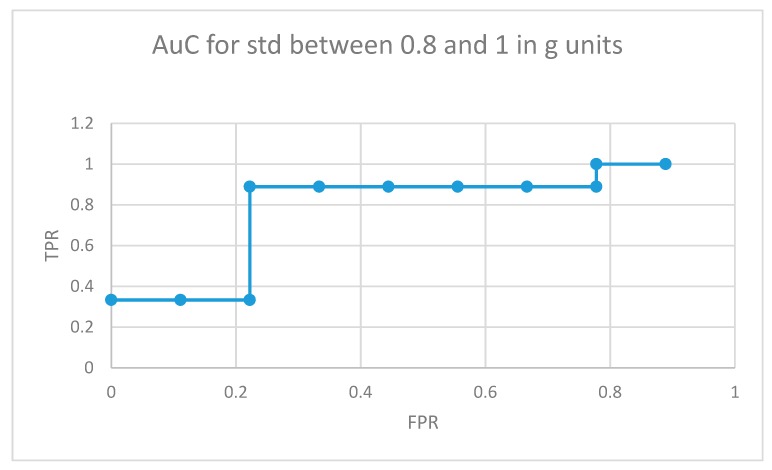
AuC for non-medicated participants with ADHD for movement fragments with the combined standard deviation between 0.8 and 1 in g units.

**Figure 9 sensors-19-02935-f009:**
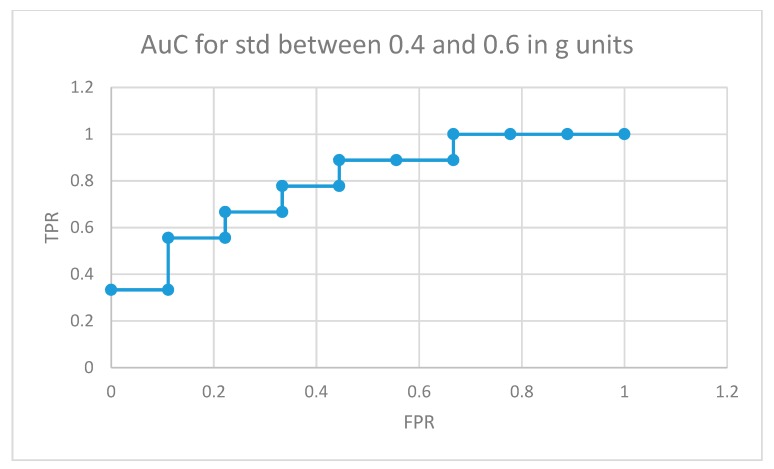
AuC for non-medicated participants with ADHD for movement fragments with the combined standard deviation between 0.4 and 0.6 in g units.

**Figure 10 sensors-19-02935-f010:**
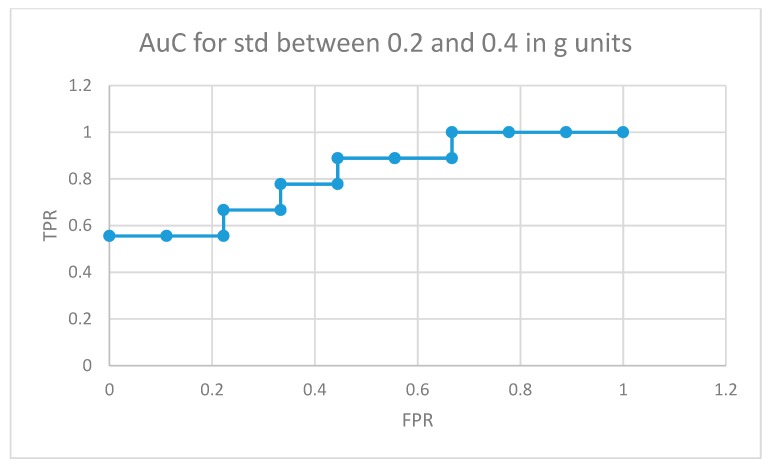
AuC for medicated participants with ADHD for movement fragments with the combined standard deviation between 0.2 and 0.4 in g units.

**Figure 11 sensors-19-02935-f011:**
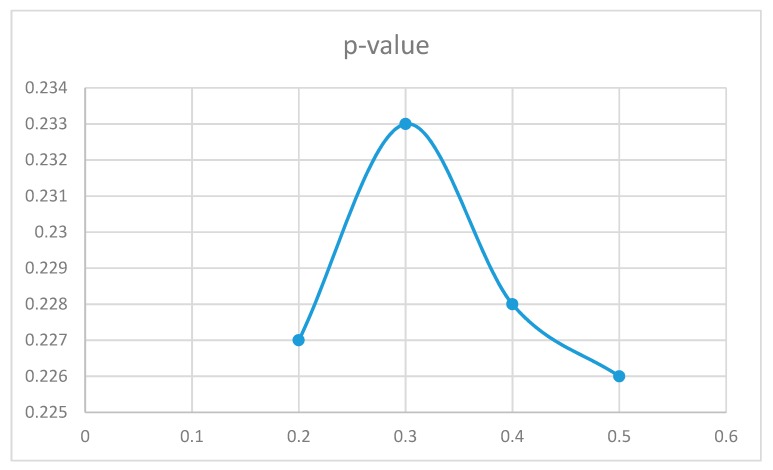
p-Values for different th values for the case of ADHD non-medicated participants with movements in the interval of 0.4 to 0.6 g.

**Table 1 sensors-19-02935-t001:** Participant demographics for ADHD diagnosed participants.

ID	Age	Gender	On Medication
ADHD 1	12	Male	No
ADHD 2	12	Male	Yes
ADHD 3	7	Male	Yes
ADHD 4	6	Male	No
ADHD 5	6	Male	No
ADHD 6	7	Male	Yes
ADHD 7	9	Male	No
ADHD 8	7	Female	Yes
ADHD 9	12	Male	Yes
ADHD 10	9	Male	Yes
ADHD 11	8	Male	Yes
ADHD 12	8	Male	Yes
ADHD 13	7	Male	No
ADHD 14	8	Male	No
ADHD 15	9	Male	Yes
ADHD 16	10	Female	No
ADHD 17	6	Female	No
ADHD 18	8	Male	No

**Table 2 sensors-19-02935-t002:** Participant demographics for control participants.

ID	Age	Gender
Control 1	12	Male
Control 2	10	Male
Control 3	12	Female
Control 4	9	Female
Control 5	7	Male
Control 6	9	Male
Control 7	12	Male
Control 8	11	Male
Control 9	16	Female
Control 10	10	Male
Control 11	12	Female
Control 12	6	Female
Control 13	9	Male
Control 14	10	Male
Control 15	12	Female
Control 16	16	Female
Control 17	6	Male
Control 18	8	Female

**Table 3 sensors-19-02935-t003:** Total number of non-similar movement fragments per group.

Group	Number of Non-Similar Fragments	Average Number Per Participant	Standard Deviation
Controls	9172	1019	484
ADHD medicated	11829	1314	676
ADHD non-medicated	13328	1481	660

**Table 4 sensors-19-02935-t004:** p-Values for the analysis of each ADHD group compared with the control group.

Group	t-Test p-Value
ADHD medicated	0.152
ADHD non-medicated	0.055

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
