# Peer review of "Using Recurrent Neural Networks to Compare Movement Patterns in ADHD and Normally Developing Children Based on Acceleration Signals from the Wrist and Ankle"

_sensors, 2019, doi:10.3390/s19132935_

Round 1

Reviewer 1 Report

The paper presents a new Recurrent Neural Network to assess differences in the movement patterns of children with ADHD. Each child is monitored for 24 consecutive hours, in a normal 21 school day, wearing 4 tri-axial accelerometers

The following aspects must be clarified or explained more clearly:

-please explain why is a new RNN - explain more clearly the differences with other networks of this type

-lines 268-269 - please explain how are chosen values for we have used: 50 memory units for the first layer LSTM cells and 20 memory units for the cells in the second layer.

-Line 169: 36 participants but table 1 contains description for 30 participants and table 2 for 28 participants - please explain these differences 

- it is not clearly what are the common patterns detected with this method - please explain. Also the number of children used for tests is small - is this number enough to validate the solution based on the extracted patterns?

-please add comparison with other state of the art method

Author Response

All the comments have been fully reviewed and the proposed changes fully and thoroughly implemented. We would like to thank the reviewers for the constructive comments which have helped us to improve the manuscript.

Reviewer 1:

The paper presents a new Recurrent Neural Network to assess differences in the movement patterns of children with ADHD. Each child is monitored for 24 consecutive hours, in a normal school day, wearing 4 tri-axial accelerometers

The following aspects must be clarified or explained more clearly:

Comment:

-please explain why is a new RNN - explain more clearly the differences with other networks of this type

Response:

The reviewer raises a good point, a better justification of the proposed architecture as compared to previous architectures has been added to the paper. We have added the following text at the end of the “related work” section as suggested:

The authors in [22] not only showed that deep RNNs achieved better results for Human Activity Recognition than other Deep Networks such as Convolutional Neural Networks (CNN) and other shallow methods such as Support Vector Machines (SVM) but also showed that the architecture of the RNN will have an impact in the results depending on the dataset. The number of layers and the size of the memory cells must consider not only the complexity of the activities to be recognized but also the size of the dataset in order to avoid overfitting issues. The study in [22] achieved optimal results for RNN structures stacking 2, 3 or 4 layers depending on the dataset. The authors in [29] achieved optimal results when stacking 3 layers. The structure proposed in this paper is adapted to the dataset that we have generated in the experimental part of the study.                

A new reference has been added to support that information:

29      M. Inoue,  S. Inoue,  & T. Nishida, “Deep recurrent neural network for mobile human activity recognition with high throughput”. In Artificial Life and Robotics, vol. 23 (no. 2) (2018), pp. 173-185.

The following text has been added to the section 3.5 to better capture the design issues as pointed out by the reviewer:

This architecture is adapted to the particular dataset generated for the current study. The number of memory cells in the first layer is selected based on the number of acceleration samples used as inputs and the number of the memory cells in the second layer is adapted to the length of the sequence to be reconstructed as an output for the similarity computation. Adding more layers or memory cells did not provide better results but created overfitting issues.

Comment:

-lines 268-269 - please explain how are chosen values for we have used: 50 memory units for the first layer LSTM cells and 20 memory units for the cells in the second layer.

Response:

The following text has been added to the section 3.5 to better capture the design issues as pointed out by the reviewer:

This architecture is adapted to the particular dataset generated for the current study. The number of memory cells in the first layer is selected based on the number of acceleration samples used as inputs and the number of the memory cells in the second layer is adapted to the length of the sequence to be reconstructed as an output for the similarity computation. Adding more layers or memory cells did not provide better results but created overfitting issues.

The number of input and output values was already captured in the paper and has not been re-added in the previous sentence but could be done if considered more appropriate. The number of memory cells are mentioned just before the previous added text.

Comment:

-Line 169: 36 participants but table 1 contains description for 30 participants and table 2 for 28 participants - please explain these differences 

Response:

Tables 1 and 2 included internal identifiers in the conducted study which were of no relevance to the reader. We have updated the identifiers so that now it is clearer that both tables contain 18 participants (for a balanced experiment). Table 1 also contains if the participant was on medication during the study or not (9 each for a balanced experiment). Both tables are now updated as suggested. We have also updated other related text in the document for consistency.

Comment:

- it is not clearly what are the common patterns detected with this method - please explain.

Response:

The following text has been added to section 3.5 to better explain what the common patterns detected are:

Since the use of the mean square error will equally weight the reconstruction error from each data segment in the training set (as added to the total error to be minimized in the training phase), those data segments which are most frequently repeated in the acceleration sequences from the training users will have a bigger influence in the adaptation of the internal parameters of the algorithm in order to better predict the upcoming values for these sequences. These most common sequences will therefore capture the movement patterns better charactering the control users in the training of the algorithm.

Comment:

Also the number of children used for tests is small - is this number enough to validate the solution based on the extracted patterns?

Response:

Some previous studies are based on less users (the study in [15] used 17 participants while we have used 36). Moreover, we have added the following text to the conclusions to recognize the limits of the study and how it will be improved in the future:

As a future work, a deeper analysis of finding common patterns in ADHD diagnosed children will be done. The effect of medication on behavioral changes over these patterns will also be studied. A more exhaustive experiment including a higher number of participants is also expected.

Comment:

-please add comparison with other state of the art method

Response:

We have completed the comparison with the previous related (most similar) study found in literature adding the following information to the conclusions section to justify the main differences and contributions in our proposal:

Previous similar research studies such as [15] also showed that for some particular activities there were statistically significant differences in the regions of the moderate intensity movements (0.5-0.8 g in that particular case) for the upper limb movements, although the authors do not perform an analysis for non-medicated vs medicated participants. In particular, the study in [15] showed bigger statistical differences in the acceleration values of the upper limbs between children diagnosed with ADHD and normally developing children while attending an art class than for math’s or language classes. Our study has extended the conducted activities to a complete 24-hour period and divided the participants diagnosed with ADHD into medicated and non-medicated cases. The authors in [15] only analyzed raw data values while our study has proposed a state-of-the-art architecture based on RNN to better capture time dependencies and patterns. Finally, the study in [15] only used 17 participants (10 diagnosed with ADHD, 7 normally developing cases) while we have incorporated 36 participants (18 diagnosed with ADHD, 18 normally developing cases).

Reviewer 2 Report

This paper presents an interesting comparative study on the movement patterns between normally developing and ADHD (medicated and non-medicated) children. The study is based on data collected from accelerometers placed on the child’s wrist and ankle during a 24 h monitoring period. A Recurrent Neural Network was trained to detect acceleration segments characterizing ADHD patterns. Results conducted with a population of 36 children can be summarized as follows: (1) medium intensity movements are similar between normally developing and medicated ADHD children and different for non-medicated ADHD children and (2) low intensity movements are different between normally developing and medicated children.

Overall, this paper addresses a novel and interesting topic on sensor data processing for biomedical applications. The paper is well written, it is easy to read and to understand. All concepts are clearly presented and results are comprehensively described. I have a positive impression of this work and I recommend its acceptance with the following minor remarks:

1. Title, May I suggest a shorter one: 

Using Recurrent Neural Networks to compare movement patterns in ADHD and normally developing children based on acceleration signals from the wrist and ankle

(wrist and ankle in singular)

2. For the sake of comparison, figures 6 and 7 should be merged. Also merge figures 8, 9 and 10

3. Typos:

3.1.  Line 82: "The first section, this section presents"--> The first section presents

3.2. Lines 114 and 301: 2 --> two. (114): The authors used two inertial…   (301): Two LSTM…

3.3 LSTM is defined in the keywords section. However, it is necessary to recall it in line 267. Line 415 is too far…

Author Response

All the comments have been fully reviewed and the proposed changes fully and thoroughly implemented. We would like to thank the reviewers for the constructive comments which have helped us to improve the manuscript.

Reviewer 2:

This paper presents an interesting comparative study on the movement patterns between normally developing and ADHD (medicated and non-medicated) children. The study is based on data collected from accelerometers placed on the child’s wrist and ankle during a 24 h monitoring period. A Recurrent Neural Network was trained to detect acceleration segments characterizing ADHD patterns. Results conducted with a population of 36 children can be summarized as follows: (1) medium intensity movements are similar between normally developing and medicated ADHD children and different for non-medicated ADHD children and (2) low intensity movements are different between normally developing and medicated children.

Overall, this paper addresses a novel and interesting topic on sensor data processing for biomedical applications. The paper is well written, it is easy to read and to understand. All concepts are clearly presented and results are comprehensively described. I have a positive impression of this work and I recommend its acceptance with the following minor remarks:

Comments:

Comment:

1. Title, May I suggest a shorter one: 

Using Recurrent Neural Networks to compare movement patterns in ADHD and normally developing children based on acceleration signals from the wrist and ankle

(wrist and ankle in singular)

Response:

We have updated the title to the one proposed.

Comment:

2. For the sake of comparison, figures 6 and 7 should be merged. Also merge figures 8, 9 and 10

Response:

We have tried to implement these recommendations, but some values where not properly read when combining the figures 6 and 7 using the same axis. We have kept the original figures for a better reading of the information although the reviewer is right that a single figure with the same axis will show the difference in values from both groups. Figures 8, 9 and 10 seem also a bit more clear if kept in different figures, but could be merged if needed.

Comment:

3. Typos:

3.1.  Line 82: "            , this section presents"--> The first section presents

3.2. Lines 114 and 301: 2 --> two. (114): The authors used two inertial…   (301): Two LSTM…

3.3 LSTM is defined in the keywords section. However, it is necessary to recall it in line 267. Line 415 is too far

Response:

All the proposed changes have been implemented as suggested.

Round 2

Reviewer 1 Report

Thank you for your updates.

All my comments were addressed.

However, regarding the comment: please explain how are chosen values for we have used: 50 memory units for the first layer LSTM cells and 20 memory units for the cells in the second layer.

line 283: "Adding more layers or memory cells did not provide better results but created overfitting issues."

It would be useful if you present some graphs with errors in order to sustain your suggestion. 

Author Response

Response to reviewers’ comments

All the comments have been fully reviewed and the proposed changes fully and thoroughly implemented. We would like to thank the reviewers for the constructive comments which have helped us to improve the manuscript.

Reviewer 1:

Thank you for your updates.

All my comments were addressed.

Comment:

However, regarding the comment: please explain how are chosen values for we have used: 50 memory units for the first layer LSTM cells and 20 memory units for the cells in the second layer.

line 283: "Adding more layers or memory cells did not provide better results but created overfitting issues."

It would be useful if you present some graphs with errors in order to sustain your suggestion. 

Response:

We have added the following text to section 4.1 with numerical results as suggested:

Changing the parameters in the architecture presented in section 3.5 (number of layers and memory cells per layer) provided similar results but required more time for training and was prone to overfitting problems. A 3-layer RNN architecture with 50 memory cells in all the 3 layers achieved a p-value of 0.054 but required 73% more time to train and the difference of non-similar segments between the control group used for training and the control group used for validation was 52% (a bigger number of parameters to be trained in the model will try to reduce the errors in the training set without guaranteeing the generalization to other control users not used in the training phase).